Some like it hot: a differential response to changing temperatures by the malaria vectors Anopheles funestus and An. gambiae s.l.

Charlwood Jacques Derek jdcharlwood@gmail.com
Instituto de Higiene e Medicina Tropical, IHMT, Universidade Nova de Lisboa, UNL, Rua da Junqueira, Global Health and Tropical Medicine, GHTM , Lisbon , Portugal
MOZDAN (Mozambican-Danish Rural Malaria Initiative) , Morrumbene , Inhambane Province , Mozambique
Centre for Health Research and Development, University of Copenhagen , Copenhagen , Denmark
National Institute of Health , Maputo , Mozambique
Braga Erika
Electronic publication date: 2017 Mar 28
Publication date: 2017
Volume: 5
Electronic Location ID: e3099
Received 2016 Apr 29; Accepted 2017 Feb 16
Copyright: ©2017 Charlwood
Copyright year: 2017
Copyright holder: Charlwood
License: This is an open access article distributed under the terms of the Creative Commons Attribution License, which permits unrestricted use, distribution, reproduction and adaptation in any medium and for any purpose provided that it is properly attributed. For attribution, the original author(s), title, publication source (PeerJ) and either DOI or URL of the article must be cited.
License URL: https://creativecommons.org/licenses/by/4.0/

Keywords: Anopheles gambiae s.l., Temperature, Anopheles funestus, Population dynamics, Rainfall, Mozambique, Light-trap, Exit collection, Gravid, Unfed

Funding: Danish Bilharziasis Laboratory, Denmark This work was funded by the Danish Bilharziasis Laboratory, Denmark. The funders had no role in study design, data collection and analysis, decision to publish, or preparation of the manuscript.

==============================
Background

With the possible implications of global warming, the effect of temperature on the dynamics of malaria vectors in Africa has become a subject of increasing interest. Information from the field is, however, relatively sparse. We describe the effect of ambient temperature over a five-year period on the dynamics of An. funestus and An. gambiae s.l., collected from a single village in southern Mozambique where temperatures varied from a night-time minimum of 6 °C in the cool season to a daytime maximum of 35 °C in the hot season.

Results

Mean daily air temperatures varied from 34 °C to 20 °C and soil temperatures varied from 26 °C to 12 °C. Diurnal variation was greatest in the cooler months of the year and were greater in air temperatures than soil temperatures. During the study 301, 705 female An. funestus were collected in 6,043 light-trap collections, 161, 466 in 7,397 exit collections and 16, 995 in 1,315 resting collections. The equivalent numbers for An. gambiae s.l. are 72,475 in light-traps, 33, 868 in exit collections and 5,333 from indoor resting collections. Numbers of mosquito were greatest in the warmer months. Numbers of An. gambiae s.l. went through a one hundredfold change (from a mean of 0.14 mosquitoes a night to 14) whereas numbers of An. funestus merely doubled (from a mean of 20 to 40 a night). The highest environmental correlations and mosquito numbers were between mean air temperature (r2 = 0.52 for An. funestus and 0.77 for An. gambiae s.l.). Numbers of mosquito collected were not related to rainfall with lags of up to four weeks. Numbers of both gravid and unfed An. gambiae complex females in exit collections continued to increase at all temperatures recorded but gravid females of An. funestus decreased at temperatures above 28 °C. Overall the numbers of gravid and unfed An. funestus collected in exit collections were not correlated (p = 0.07). For an unknown reason the number of An. gambiae s.l. fell below monitoring thresholds during the study.

Conclusions

Mean air temperature was the most important environmental parameter affecting both vectors in this part of Mozambique. Numbers of An. gambiae s.l. increased at all temperatures recorded whilst An. funestus appeared to be adversely affected by temperatures of 28 °C and above. These differences may influence the distribution of the vectors as the planet warms.

Introduction

Temperature is a major driving force in insect populations. In the laboratory the species that are almost entirely responsible for malaria transmission in Sub-Saharan Africa, the freshwater members of the Anopheles gambiae complex (An. gambiae, An. coluzzi and An. arabiensis) and Anopheles funestus, respond differently to different temperature regimes. Larval development rates for An. arabiensis peak within the temperature range of 22–32 °C with survival rates also highest at 32 °C, whilst an optimal temperature for larval development of An. gambiae is between 28 and 32 °C but survival rate to adulthood is highest between 22–26 °C (Bayoh & Lindsay, 2003, Christiansen-Jucht et al., 2014). This reflects the higher temperature tolerance of An. arabiensis compared to An. gambiae (Kirby & Lindsay, 2004) which itself may be responsible for the extended distribution of the former species into hotter and drier environments in Africa. Anopheles funestus, on the other hand, has a single optimum temperature of 25 °C for development with substantial declines in survival either side of this (Christiansen-Jucht et al., 2014). Le Sueur & Sharp (1991) concluded that the effect of temperature on An. merus (another member of the An. gambiae complex) was greatest during metamorphisis in the pupal stage, as did Heuval van den (1963) for Aedes aegypti. Temperature-related metabolism during metamorphosis was affected by available energy reserves (gathered during larval development) (Le Sueur & Sharp, 1991). At higher temperatures reserves are lower than at lower temperatures, hence emerging adults are smaller. In the laboratory none of the An. gambiae complex or An funestus survived as larvae or pupae at temperatures above 35 °C. In the field, however, larvae and pupae of An. gambiae s.l. (probably An. arabiensis) have been found in pools at temperatures of 40.5–41.8 °C (Holstein, 1952, quoted in Gillies & De Mellion, 1968).

The differential response to temperature in the laboratory reflects the different kinds of water in which larval development takes place in the field. Thus, although immature forms of the An. gambiae complex may occur in a great variety of water bodies, the most characteristic are the ‘shallow open sun-lit pools with which every field worker in Africa is familiar’ (Gillies & De Mellion, 1968). Immatures of An. funestus are generally found in more permanent, shaded, water bodies with emergent vegetation that are cooler than exposed puddles.

Diurnal fluctuations in temperature also affect the development of many insects (Vangansbeke et al., 2015). In addition to being hotter, temperature fluctuations in small pools are greater than those in larger, shaded, bodies of water. For example, although temperature minima in pools typically used by An. gambiae are similar to the minima in shaded ones, maxima may be 10 °C higher (Haddow, 1943) and, not surprisingly, given their larval habitat, both An. gambiae and An. arabiensis, also respond better to fluctuating temperatures than do An. funestus (Lyons, Coetzee & Chown, 2013; Christiansen-Jucht et al., 2014).

In contrast to the larvae, adults of both An. gambiae s.l. and An. funestus, experience similar microclimates due to their predominantly endophilic behaviour. Temperature influences the time it takes for egg development following a blood-meal but may also have subtler effects. For example, An. funestus delays returning to feed following oviposition at temperatures above 26.5 °C, but at lower temperatures females re-feed shortly after egg laying (Gillies & Wilkes, 1963). Ironically, the extra time spent in returning to feed at higher temperatures is compensated for by it taking two rather than three days for the mosquito to complete egg development, so that the duration of the complete gonotrophic cycle is three days at all temperatures (Gillies & Wilkes, 1963).

The effect of temperature on the larval stages is manifest in the emergent population (Beck-Johnson et al., 2013). Newly emerged insects can be distinguished from the mature population by their abdominal and gonotrophic state. Newly emerged males have un-rotated terminalia (Charlwood, 2011) whilst females have undeveloped ovaries and constitute the unfed portion of the resting or exiting population from houses (Gillies & Wilkes, 1965; Charlwood, Thompson & Madsen, 2003). The relative proportions of engorged to gravid females at different temperatures also provides information on the duration of egg development in mature insects.

Surprisingly, there remains a lack of comprehensive data on the effects of temperature and other environmental factors on mosquito population dynamics in the wild. Possible effects of temperature on mosquitoes in the field are most easily observed in areas with a wide variation in both daily and seasonal temperatures. Wild mosquito populations are, however, notoriously unpredictable and short-term, chaotic, fluctuations are common. Long-term observations can assist in reducing the ‘noise’ in such data. Here we describe the effect of ambient temperature, and other environmental parameters, over a five-year period, on the dynamics of newly emerged and mature An. funestus and An. gambiae s.l. Insects were collected from a single village in southern Mozambique where temperatures varied from a night-time minimum of 6 °C in the cool season (10.5 °C below the lower limit of 16.5 °C for larval activity, (Jepson, Moutia & Courtis, 1947)) to a daytime maximum of 35 °C in the hot season.

Methods

Description of study site

The approximately 5 × 4 km village of Furvela, (23°43′S, 35°18′E), 475 km north of the capital Maputo, is bordered on two sides by the alluvial plain of two river systems (Fig. 1). The Furvela River valley to the north of the village in particular has a considerable amount of local irrigation for agriculture, which provides a large and relatively stable number of small canals. The Inhnanombe river to the east of the village consists largely of beds of the reed (caniço), used for housing, and sugar cane, used in the production of local alcohol; it does not flow as fast as the Furvela river. Anopheles funestus predominates on the Furvela River side of the village and An. gambiae s.l. on the Inhnanombe side (Kampango et al., 2013). Malaria is endemic in the village with 80% of 2–4 year olds being positive in a cross sectional survey undertaken in 2006 whilst children under one year of age comprised the majority of attendees at a clinic established by the project in 2001 (Files S1 and S2).

Figure 1 Google Earth image of Furvela Village.

Furvela Village and its environ© Google Earth; Image 2016 ©CNES/ Astrium. Note the location of Linga Linga where information on the malaria and mosquitoes is also available (Charlwood et al., 2013; Charlwood et al., 2015).

Houses in the village are generally made with caniço walls and palm thatch roofs. Although most houses don’t have windows the majority have a ca. 15 cm gap between the roof and walls at either end of the house. Doors and doorframes are also generally badly fitting; hence mosquitoes can easily enter these houses. Other styles of house include those with corrugated iron sheets for the roof and those made of concrete blocks (which do have windows). Houses are built either in family compounds of three to six houses or as relatively evenly spaced individual homes.

At the start of the study houses were mapped with handheld Global Positioning Units (Garmin etrex), numbered and the manner of construction and size noted (Files S3 and S4). A census was taken and residents were informed about the purpose of the study and consent concerning the possibility of future mosquito collections obtained. The initial study area was expanded in the second year of the study and mapping was also repeated in 2007.

Mosquito Collection

Light-traps

Host seeking mosquitoes were collected in CDC light-traps hung, inside bedrooms, approximately 1.5 m from the floor at the foot of the bed of people who themselves were sleeping under mosquito nets. A random list of houses was produced (using the random number generator in Microsoft Excel) and routine collections were made according to the list. In addition, a number of houses known to have high densities of mosquitoes were used as sentinel sites. Collections were made in 764 houses on the Furvela river side of the village and 214 on the Inhnanombe side of the village. Eleven houses were used for sentinel collections; each being sampled for more than 100 nights.

Exit collection

From 2003 to 2007 mosquitoes were also collected exiting houses at dusk (Charlwood, 2011). The door of the house was left open and covered with a white mosquito-netting curtain. Mosquitoes were manually aspirated off the curtain as they attempted to leave. See: https://www.youtube.com/watch?v=SL8FeIuY1GM.

Most of the houses used for the exit collections were on the Furvela side of the village. The young men and women who did the work, and who lived in dispersed locations collected from their own or nearby houses. Altogether collections were made from 501 houses some of which also acted as sentinel sites.

Resting collection

Resting collections, using a torch and an aspirator, were performed on the Furvela side of the village, on an ad hoc basis on 163 days, from a total of 132 houses (mean number of 3.9 collections per house) where mosquito nets were not in use, and, initially, outdoors.

Mosquito processing

Collected Anopheles were separated into species or species group, according to the keys of Gillies & De Mellion (1968) and Gillies & Coetzee (1987) and sexed. Females were further separated into unfed, part-fed, engorged, semi-gravid and gravid categories. A number of the An. gambiae s.l. were identified to species by PCR. DNA extraction was performed individually following the protocols of Collins et al. (1988) and the species identified using the protocols of Scott, Brogdonw & Collins (1993). A small number of An. funestus were also identified by PCR using the protocols of Koekemoer et al. (2002).

Meteorology

Temperature, insolation and windspeed measurement

Daily variation estimates of soil and air temperature, insolation and windspeed were obtained with a Delta-T digital weather station (Delta-T Devices, Cambridge, UK) that recorded hourly information at the edge of the village. Soil temperatures approximate those that larvae are exposed to whilst air temperatures are those that more closely approximate those that adults may be exposed to. Unfed females exiting houses at dusk are newly emerged (Charlwood, Thompson & Madsen, 2003) and reflect the effect of temperature on the larvae whilst the ratio of unfed to gravid insects may reflect temperature effects on the adults. Unfortunately, the weather station did not operate throughout the study, nor did the humidity or rainfall meter work consistently. The longest hourly data sets were from 3rd May 2004 to 1 October 2005 and from 10 Nov 2007 to 24 Nov 2008. Hourly data from all years, including the later ones, was amalgamated into daily data and daily data amalgamated into ISO weeks. Mean values for the different ISO weeks from all the weather station files were determined and used in calculations (File S5).

Long term temperature data recorded at Vilanculos, a town ∼200 km north of Furvela, were also obtained (long-term data available from http://www.tutiempo.net).

Rainfall data

Rainfall data was available from the town of Maxixe, 20 km to the south of Furvela. Since the distribution of rainfall is important (20 mm falling on seven consecutive days in a week is likely to have a different effect than 140 mm falling on a single day) a modified measure of rainfall was used to estimate effects:

• Modified weekly rain = (Rain (mm) * #rain days)/7

• Thus 140 mm on a single day is equivalent to (140∗1)∕7 = 20

• And 140 mm with rain every day is equivalent to (140∗7)∕7 = 140.

The daily, weekly and monthly records of rainfall over the period 2000–2010 are available at File S6 (Rainfall data).

Analysis

Data were entered into, and analyzed with, Excel (File S6). Unfed mosquitoes from light-trap collections represent all age groups. Unfed mosquitoes from exit collection are, however, almost entirely newly emerged ones (Charlwood, 2011), whilst gravid females have taken at least one blood meal sufficient to develop eggs, and will also include infectious ones. The weekly Williams mean (log10 (n + 1)) of these three groups of An. funestus and An. gambiae s.l. were compared to mean, maximum and minimum temperatures, temperature difference, insolation and wind speed, measured in Furvela, and modified rainfall measured in Maxixe.

The relationship between mosquito numbers and environmental factors was examined using bivariate correlations, and Pearson’s correlation coefficient (2-tailed P value ≤ 0.05 significance). Least squares multiple linear regression (with climatic factors as independent variables) was also undertaken using the Excel add-in StatPlus. The most parsimonious model was determined by subtraction of least important variables.

Ethics

The study was conducted under the aegis of the joint Instituto Nacional de Saúde (INS)—DBL Centre for Health Research and Development project ‘Turning houses into traps for mosquitoes’, which obtained ethical clearance from the National Bioethics Committee of Mozambique on 2 April 2001 (ref: 056/CNBS/01). Householders were informed about the purpose of the collections. Verbal consent was obtained when collections were initiated.

Results

Environmental variables

Mean temperatures recorded at Vilanculos were higher than those recorded in Furvela, but both followed a similar pattern (File S7 Temperature data). There was both a marked seasonality in temperatures and considerable variation from one day to the next. Mean soil and air temperatures from Furvela, derived from hourly measurements, 10 Nov 2007–24 Nov 2008 are shown in Fig. 2. Diurnal variation in temperature differed between cool and hot seasons. Figure 3A shows the diurnal pattern recorded on the three coolest nights of the year (16–18th July) and 3B the three warmest (14–16th October). Overall at the higher temperatures daily variation (difference between maximuim and minimum temperature) was less than it was at the cooler temperatures (Fig. 4). At a mean of 23.7 °C the variation in air temperatures was 5.9 degrees and at 18.1 °C was 15.7 degrees. Variation is soil temperature was lower than air temperatures being 4.1 degrees at 30.3 °C and 8.9 degrees at 28.2 °C.

Figure 2 Annual temperature variation.

Mean daily soil and air temperatures recorded by the Delta logger in Furvela village.

Except for the mornings mean soil temperatures were consistently circa 5 °C warmer than air temperatures. Mean amounts of insolation showed a similar pattern to temperature.

Figure 3 Daily variation in temperature during (A) the coolest months of the year and (B) the warmest months of the year, Furvela, Mozambique

Figure 4 Difference between maximum and minimum air and soil temperature recorded from Furvela village, Mozambique, by ISO week number.

Mosquito data

Twenty-six (86%) of 30 males and 331 (81.3%) of the 407 females from an unselected sample of the An. gambiae complex identified by PCR from 2002 and 2004 comprised An. gambiae, the other species being 4 male and 67 (16.5%) female An. arabiensis and nine (2.2%) female An.  merus (Table 1). There was no statistical difference in the ratio of An. gambiae and An. arabiensis according to method of collection (light-trap, exit collection or resting collection) (File S8).

Table 1 PCR identifications of members of the An. gambiae complex collected from light-traps, Furvela Village, Mozambique.

Year	2002	2003	2004	Total	
Species	N	%	N	%	N	%	N	%	
A. arabiensis	16	27.1	35	23.3	20	8.8	71	16.2	
A. gambiae	43	72.9	108	72.0	206	90.4	357	81.7	
A. merus	0	0.0	7	4.7	2	0.9	9	2.1	
Total per year	59	100.0	150	100.0	228	100.0	437	100.0	

All of An. funestus examined morphologically had a single pale spot on the upper branch of the 5th vein and did not have a pale spot at the tip of the 6th vein and corresponded to An. funestus. Seventy-one females of the An. funestus group were identified by PCR (Fil S9, courtesy of A. Szlanski). All were An. funestus. Given that this is the endophilic member of the species group, and that it was endophilic behavior that was studied, it is assumed that this was the only member of the species group present in our collections.

301, 705 female An. funestus were collected in 6,043 light-trap collections, 161, 466 in 7,397 exit collections and 16, 995 in 1,315 resting collections. The equivalent numbers for An. gambiae s.l. are 72, 475 in light-traps, 33, 868 in exit collections and 5, 333 from indoor resting collections (File S10—Raw data). Outdoor resting collections failed to produce any mosquitoes. Other anopheline species collected in light traps included 5776 Anopheles tenebrosus, 725 Anopheles letabensis, 22 Anopheles rufipes, five Anopheles squamosus, and a single Anopheles pharoensis. A further 219 An. tenebrosus and five An. rufipes were collected exiting houses.

Figure 5 shows the weekly mean numbers collected per house per night of An. funestus and An. gambiae s.l. in light traps and exit collections over the course of the study in conjunction with temperatures recorded at Vilanculos and modified rainfall from Maxixe. Over the three years when both light trapping and exit collections were simultaneously undertaken (2003–2006) mean numbers of An. funestus per house, per method, were similar. In 2007 a cordon sanitaire of long lasting insecticide nets (LLIN’s) was established around the Furvela River valley (JD Charlwood, 2007, unpublished data) and numbers in exit collections decreased relative to numbers in light-traps. Numbers of An. gambiae s.l. in exit and light-trap were also similar. Over the course of the project, however, An. gambiae s.l. disappeared from both light-trap and exit collection collections. Given the possible effect of the cordon sanitaire on numbers collected further analysis is confined to the years 2001–2006 (when 5,090 light-trap, 4,461 exit and 1,315 resting collections were performed).

Figure 5 Temperature, rainfall and number of mosquitoes collected in Furvela.

Upper graph—Rainfall (measured in Maxixe) (Blue histogram), air temperature (measured in Vilanculos) (orange line) (A) mean numbers of unfed Anopheles funestus form light-traps (blue line) and in exit collections (green line) (B) mean numbers of unfed Anopheles gambiae collected from light-traps (Dark blue line) and in exit collections (light blue line) from Furvela village, 2001–2009. Note the log scale.

Numbers of mosquito were greatest in the warmer months. Figure 6 shows the mean adjusted rainfall from Maxixe, mean soil and air temperature and wind speed from Furvela, mean numbers of unfed An. funestus and An. gambiae s.l. collected in light-traps and mean numbers of unfed and gravid insects by species from exit collections by ISO week in the years 2001–2006. Unfed insects in exit collections comprised the newly emerged portion of the population. Gravid insects represent the gonotrophically active (mature) population. The proportion of that population that they represent depends on the duration of oogenesis following a blood meal. At warmer temperatures development takes two days and so the gravid sample represents half of the mature population whereas at cooler temperatures development may take three days and so the gravid sample will only represent one third of the mature population. The whole of the newly emerged population, on the other hand, is sampled every day.

Figure 6 Weekly mean adjusted rain, soil temperature, air temperature, windspeed and mosquito numbers.

Environmental parameters and mosquito numbers by ISO week, Furvela, Mozambique. (A) mean weekly adjusted rain in millimetres (measured in Maxixe), (B) Light blue line —mean soil temperature (in degrees Centigrade); Black solid line Mean air temperature (in degrees centigrade); dotted line —mean wind-speed (in metres per second), (C) Mean number of Anopheles funestus (red line) and An. gambiae s.l. (blue line) collected from light traps, (D) Mean number of unfed (pink line) and gravid (red line) An. funestus from exit collections,(note the log scale) (E) Mean number of unfed (light blue line) and gravid (dark blue line) An. gambiae s.l. from exit collections,(note the log scale).

The maximum Pearson correlation co-efficients between mosquito numbers and individual environmental parameters are shown in Table 2. Table 2 also gives the correlation co-efficients between the different mosquito groups. The highest environmental correlations and mosquito numbers were between temperature and all collections of An. gambiae s.l. (see File S11 for the other possible correlations). Adding a lag of up to four weeks to the light-trap data from either species did not improve the Pearson correlation co-efficient between rainfall and numbers (Table 3) Correlation co-efficients were always lower in comparisons involving An. funestus.

Table 2 Correlation coefficients between mosquitoes and climate.

Correlation coefficients between weekly mean numbers of mosquitoes according to the collection method and the highest correlation co-efficient by environmental variable, Furvela, Mozambique 2003–2006.

Species	Sample	Environmental variable	Correlation co-efficient	p	
Anopheles funestus	Light-trap	Air temperature	0.5352	>0.0005	
	Unfed exit	Min air temperature	0.71959	>0.0005	
	Gravid exit	Max Solar	0.68915	>0.0005	
	Unfed exit	Gravid exit	0.25169	0.071	
Anopheles gambiae s.l.	Light-trap	Air temperature	0.75105	>0.0005	
	Unfed exit	Air temperature	0.73002	>0.0005	
	Gravid exit	Air temperature	0.74211	>0.0005	
	Unfed exit	Gravid exit	0.86102	>0.0005	
funestus/gambiae	Unfed exit	Unfed exit	0.58979	>0.0005	
	Gravid exit	Gravid exit	0.44756	0.0009	

Table 3 Correlation between mosquitoes and rainfall.

Pearson correlation co-efficients between adjusted rainfall and numbers of An. gambiae s.l. and An. funestus collected in light traps at lags of 0–4 weeks.

Lag (weeks)	An. gambiae s.l.	An. funestus	
0	0.671	0.245	
1	0.435	0.285	
2	0.517	0.275	
3	0.358	0.150	
4	0.379	0.248	

Figure 7A gives the relationship between numbers of An. funestus and An. gambiae s.l. caught in light-traps and air temperatures and Fig. 7B between mosquito numbers and soil temperature (r2 = 0.52 for An. funestus and 0.77 for An. gambiae s.l.). Although both species increased significantly as temperatures increased, numbers of An. gambiae s.l. went through a one hundredfold change (from a mean of 0.14 mosquitoes a night to 14) whereas numbers of An. funestus merely doubled (from a mean of 20 to 40 a night).

Figure 7 Mean numbers of Anopheles funestus. (red) and An. gambiae s.l. (blue) collected in light-traps from Furvela village, Mozambique (A) by air temperature and (B) by soil temperature (in degrees Centigrade).

The number of unfed and gravid insects in exit collections by mean air temperature are shown in Figs. 8A and 8B. At temperatures below 28 °C the mean number of gravid An. funestus collected increased as the temperature increased; and at a faster rate than the rate of increase in immature insects but at temperatures above 28 °C the number decreased whilst numbers of newly emerged insects continued to increase (Fig. 8A). Numbers of both gravid and unfed An. gambiae complex females in exit collections continued to increase at all temperatures recorded (Fig. 8B). The proportion of the An. gambiae s.l. population that was gravid was more variable at lower temperatures. This was probably due to the small sample sizes at these temperatures. The proportion of An. funestus on the other hand was more variable at the higher temperatures but why this should be so remains unknown since the data comes from a time when large-scale interventions had not been applied. Overall the numbers of gravid and unfed An. funestus collected in exit collections were not correlated (p = 0.07).

Figure 8 The relationship between the mean number of (A) Anopheles funestus and (B) An. gambiae s.l. in exit collections and soil temperature Furvela, Mozambique.

The best models for each species and each category of mosquito are given in Table 4. Adjusted rainfall was only included in one model. The models explained more of the variation in An. gambiae s.l. than they did for An. funestus although the environmental parameter used in the models, with the exception of the best model for An. funestus and An. gambiae s.l. in light-traps, were the same. The explanatory values (the adjusted R2) were all higher for the An. gambiae s.l. than for the An. funestus.

Table 4 Models and mosquito numbers.

Environmental regression models for the different categories of female mosquito collected, Furvela, 2001–1007.

Species	Category	Parameters	R2	p	
An. funestus	Unfed exit	Air temp + windspeed	0.611	>0.00005	
	Gravid exit	Soil temp + windspeed	0.563	>0.00005	
	Unfed Light	Air temp + Rain	0.344	>0.00005	
An. gambiae s.l.	Unfed exit	Air temp + Windspeed	0.752	>0.00005	
	Gravid exit	Soil temp + Windspeed	0.822	>0.00005	
	Unfed Light	Air temp + Windspeed	0.756	>0.00005	
Notes.

The equations for the different models are listed below:

Log An. funestus Light-trap = +0.9178−0.445*log rain + 0.0342*Air temp. Log An. gambiae s.l. Light-trap = −1.419 + 0.117*Air temp − 0.392*Windspeed.

Exit An. funesus unfed = −0.100 + 0.044*Air temp + 0.156*Windspeed Exit An. funestus gravid = −0.099 + 0.0456*Soil temp − 0.111*Windspeed.

Exit An. gambiae s.l. unfed = −1.629 + 0.1097*Air temp − 0.446*Windspeed Exit An. gambiae s.l. gravid = −2.05 + 0.10869 Soil temp − 0.734*Windspeed.

The abdominal status of mature females collected from 1315 resting collections and mean monthly temperature is shown in Fig. 9. For both, species, or species group, a higher proportion of semi-gravid and gravid females compared to engorged females were collected during the cooler months of the year (May–August). In other words, oogenesis was taking longer at the lower temperatures.

Figure 9 Proportion of Anopheles funestus and An. gambiae s.l. indoor resting that were gravid at the time of collection and mean temperature, Furvela Mozambique.

The proportion gravid to engorged An. funestus of 50% occurred at 25 °C. Thus, at these temperatures, and above it took two days to mature the ovaries and below this three days post-feeding to mature them. The proportion gravid of An. gambiae s.l. from resting catches was always lower than that of the An. funestus and only reached 50% at the highest temperatures. At mean temperatures of 21.5 °C 76% of the An. funestus collected were semi-gravid and gravid. This implies that it was taking not just three but four days to complete gonotrophic development.

Discussion

With the possible implications of Global Warming, the effect of temperature and other environmental parameters on the dynamics of malaria vectors in Africa is an area of increasing interest. The temperature in Furvela fluctuates between the minimum and optimum temperature for mosquito development, hence over the linear part of the reaction norm. Temperature was the most important environmental parameter, of those measured, determining mosquito numbers in the village. Although a more in depth assessment would probably improve the interpretation of the results, even in the straightforward analysis presented here mean daily temperatures from either air or soil sensors explained 70% and 35% of the density changes observed in An. gambiae s.l. and An. funestus respectively (Table 2). As expected, given its rapid developmental time, rates of increase were substantially higher in the An. gambiae s.l. compared to the An. funestus. Sporozoite rates were also highest during the warmer months of the year (File S12). The combined effect on the entomological infection rate (EIR) and malaria morbidity in the village will be examined elsewhere.

The population dynamics of An. funestus on the peninsula of Linga Linga across the Morrumbene Bay (Fig. 1) is determined by rainfall (Charlwood et al., 2013). The absence of a relationship between rainfall and mosquito numbers, particularly the An. gambiae s.l., in Furvela is surprising. It may be due to the sandy soil in the village which means that rain is rapidly absorbed and the puddles are rare or absent. At the same time breeding conditions for the An. funestus in the valley remain reasonably constant due to the irrigation practices of the local farmers and even major storms have only a temporary effect on the dynamics of the mosquito (Charlwood & Braganca, 2012).

The ratio of gravid to unfed mosquitoes in exit collections depends on a number of factors, in particular house construction. The two sets of females enter the house at different times (unfed newly emerged insects entering at dawn to rest and, soon-to-be-gravid, host seeking females, to feed throughout the night). They use different cues (visual contrast and odour) and enter through different routes (open doors and eaves) (Gillies, 1988). Thus, houses that may allow access for one group are not necessarily suitable for the other. In addition to house effects, the proportion of egg development time spent inside houses (which is presumed to be 100%, at least for An. funestus), the survival rate per oviposition cycle and the duration of oogenesis, can all affect the unfed/gravid ratio. Should any of these factors change with temperature then the overall ratio will also change with temperature. The absence of change, as occurred with the An. gambiae s.l. implies that these factors remained constant, or compensated exactly, over the observed range of temperatures.

Ironically, the highest correlation between gravid insects in exit collections (for both species or species group) was with soil temperature (which mimics water temperature) whilst for unfed (newly emerged) insects it was with outdoor air temperature. As pointed out by Paaijmans et al. (2010) the micro-climate experienced by the mosquito inside houses may be quite different to that outside. Houses may be warmer in the cool season and cooler in the hot season than temperatures recorded outside. Nevertheless, more sophisticated measurement would only improve models for An. gambiae by a maximum of 30% and for An. funestus by 47% (and would imply that other factors, such as humidity, were of lesser importance).

Unlike An. funestus both newly emerged and gravid An. gambiae s.l. increased in a similar fashion through the whole range of temperatures experienced in Furvela. There was no apparent effect of increasing temperatures on survival and the proportion of gravid to unfed insects remained more or less constant at all temperatures. The unfed/gravid ratio of the more common An. funestus did, however, change with temperature. As temperatures increase above 26.5 °C a higher proportion of gravid An. funestus is to be expected in exit collections since the duration of oogenesis is reduced from three to two days (Gillies & De Mellion, 1968). At the temperatures recorded in July it might take three or more days, as evidenced in the resting collections and formerly described by Gillies and Wilkes Gillies and Wilkes (1963) and Gillies and Wilkes (1965). At the higher temperatures exit collections would therefore be expected to sample one half of the mature population (the other half being the semi-gravid insects that may move from one resting site to another, but in the absence of disturbance, do not leave the house) but at cooler temperatures only one third, or even less, of the population. At the higher temperatures, however, the proportion of gravid insects in the exit collections decreased, such that overall there was no significant relationship between the numbers of gravid and unfed insects in exit collections. This either means that that survival between emergence and maturity (i.e., becoming gravid) decreases at cooler temperatures or that post-teneral insects have a higher mortality at higher temperatures. Both sets of insects leave houses at the same time (Charlwood, 2011), hence sampling efficiency for the two groups should be the same.

A reduced survival among post-teneral adult An. funestus at the higher temperatures is possible as described from the laboratory (Christiansen-Jucht et al., 2015). High temperatures, independent of humidity, can have a lethal effect because as body temperature increases, metabolism and respiration increase up to a critical thermal limit, and there is a loss of integration between protein stability and metabolic processes that leads to death. Anopheles gambiae s.l. are larger than An. funestus. Volume to surface ratios differ and this may influence the ability of the adult insects to survive higher temperatures. Larger mosquitoes have a smaller surface to volume ratio and larger water reserves, which would allow them to offset the respiratory and cuticular water loss.

With one exception, windspeed was the only environmental parameter, other than temperature, included any model. Together they explained up to 82% of the An. gambiae changes and 61% of the An. funestus. The exception was An. funestus in light traps in which adjusted rain was included. Although still significant, this model had the lowest explanatory value (34%) of all the models.

Recently Paaijmans et al. (2010) have described that in addition to mean temperatures ‘the key mosquito-related traits that combine to determine malaria transmission intensity (i.e., parasite infection, parasite growth and development, immature mosquito development and survival, length of the gonotrophic cycle, and adult survival) are all sensitive to daily variation in temperature’. Fluctuations in temperature (i.e., the difference between maximum and minimum temperatures) were greatest in the cooler months. In the cool season the observed patterns in soil and air temperature were similar to shaded and open water as determined by Haddow (1943). In the warmer months’ fluctuations in temperature were less than at lower temperatures. Fluctuations around low mean temperatures can speed up rate processes, whereas fluctuations around high mean temperatures can slow them down (Paaijmans et al., 2010). Thus, the An. gambiae s.l. were well suited to the temperature regimes experienced in Furvela.

Unfortunately, the species composition of the resting or exiting An. gambiae s.l. compared to those entering the house is not known. Nevertheless, the proportion of gravid insects in resting catches varied in a similar fashion to that seen among the An. funestus. It is also possible that the different members of the An. gambiae complex behaved differently or disappeared from the study area at different rates. Nevertheless, all members of the complex did apparently disappear during the study so that, perhaps it was not just a specific species that was affected but was a complex wide problem. Meyrowitsch et al. (2011) were unable to determine the cause of the decline of An. gambiae s.l. in the Tanga region of Tanzania, three thousand kilometers to the north of Furvela. In the Kilifi area of Kenya population decline of An. gambiae, shown by a reduction in genetic diversity in the mosquito, was attributed to the introduction of LLIN’s (O’Loughlin et al., 2016). The decline in Furvela started before the introduction of any control measures and although the introduction of LLIN’s may have exacerbated the problem for the mosquito it may not have been the cause of the decline in the first place. The decline also paralleled that observed in malaria in the Rufiji basin (Ishengoma et al., 2013). That similar declines appeared to occur over a 3,000 km stretch of coastline indicates that a climatic factor was responsible.

Conclusion

Temperature is a major driving force in the dynamics of both An. funestus and An. gambiae s.l. in Furvela. Whilst gravid An. funestus appeared to be negatively affected by the highest temperatures encountered increasing temperatures were favorable for the An. gambiae s.l. which went through hundredfold changes in densities as a result. As average temperatures increase due to Global Warming areas where An. gambiae is the vector, if there remains sufficient water available for larval development, will be at heightened risk of malaria transmission.

Supplemental Information

File S1 Data set from the 2006 parasitology survey

Dataset from the first parasitology survey—children randomly selected—data to be used in a follow up article.

Click here for additional data file.

File S2 Aid post data set from Furvela for selected years

Dataset from the Furvela Aid Post excluding monthly means.

Click here for additional data file.

File S3 Census data from 2005 including construction of house and animals

Census data from 2005 Furvela—house numbers are those used in the map database.

Click here for additional data file.

File S4 Garmin Mapsource file of house positions in Furvela

Garmin database of house positions in Furvela used to create Fig. 1.

Click here for additional data file.

File S5 Environmental data from Furvela village

Hourly data from the Delta logger in Furvela village, Mozambique.

Click here for additional data file.

File S6 Raw rainfall data from Maxixe town

Click here for additional data file.

File S7 Temperature data from Vilanculos and Inhambane

Click here for additional data file.

File S8 PCR analysis of the Anopheles gambiae s.l. identified to species during the project data kindly supplied by Dr Joao Pinto

PCR data kindly provided by Dr Joao Pinto, GHTM Universidad Nova de Lisboa, Portugal.

Click here for additional data file.

Fil S9 Haploytypes of Anopheles funestus group mosquitoes from Furvela, data kindly supplied by Prof Alan Szlanski and James Austin

Haploytypes of Anopheles funestus group mosquitoes from Furvela, data kindly supplied by Prof Alan Szlanski and James Austin.

Click here for additional data file.

File S10 Mosquito collection data from Furvela

Click here for additional data file.

File S11 Complete correlation coefficients for environmental variables and mosquito numbers

Click here for additional data file.

File S12 Sporozite ELISA data from Furvela

Sporozoite data obtained using the ELISA of A. gambiae s.l and A. funestus from Furvela for the years 2001–2010

Click here for additional data file.

Data S1 Column headings for the accompanying RAW data files

Click here for additional data file.

I would like to thank Joao Pinto of the IHMT, Portugal for identifying the members of the An. gambiae complex in this study and for his perceptive comments on the manuscript. I also thank Corey LeClair of the LSHTM UK and Olivier Briët of the Swiss TPH for comments on the manuscript. Thanks too to the reviewers of the manuscript whose careful comments improved the manuscript. Thanks to Nelson Cuamba and Ricardo Thompson for logistic support during the study and thanks to the members of the MOZDAN project and to the villagers of Furvela who let us collect mosquitoes from their houses.

Additional Information and Declarations

Competing Interests

Author Contributions

Human Ethics

Data Availability

The author declares there are no competing interests.

Jacques Derek Charlwood conceived and designed the experiments, performed the experiments, analyzed the data, wrote the paper, prepared figures and/or tables, reviewed drafts of the paper.

The following information was supplied relating to ethical approvals (i.e., approving body and any reference numbers):

Ethical clearance from the National Bioethics Committee of Mozambique on 2 April 2001 (ref: 056/CNBS/01). Householders were informed about the purpose of the collections. Verbal consent was obtained when collections were initiated.

The following information was supplied regarding data availability:

The raw data has been supplied as a Supplemental Dataset.

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
