# Peer review of "Some like it hot: a differential response to changing temperatures by the malaria vectors Anopheles funestus and An. gambiae s.l"

_PeerJ, doi:10.7717/peerj.3099_

## Round 0.1 · original submission · Minor Revisions

The manuscript consists of a considerable data set that adds an important contribution to the field. There are some points that need to be well addressed as detailed by the reviewers. Please consider to focus the introduction highlighting the hypothesis. Please, also clarify the sampling strategy adjusting the results description and consider to add a map of the studied area.

Reviewer 1 ·

Basic reporting

I found this manuscript strangely difficult to review – and I am not 100% sure why. I think one of the issues is that the Introduction is written more like a Discussion so it is a little confusing to understand what it is trying to convey – although as, in the end, it highlights the confusing situation in understanding exactly how temperature affects these vectors, perhaps it is doing a good job after all! I think the manuscript could benefit from a slightly shorter Introduction that more clearly sets out the situation, and gives fewer specific examples of specific studies done, but instead summarises them. Of course this is a writing style issue/preference rather than a mandatory edit – but, for example, the final paragraph of the Introduction seems really clear in comparison.

Experimental design

No comments

Validity of the findings

One particular worry (although I am not a statistician, so may have missed something) is that some of the analyses may not be suitable for the data type (a time series), specifically Pearson’s correlation which also assumes a linear relationship. Therefore some of the outputs and correlations may not be 100% valid. It would be useful to have some more simple scatterplots (like Fig 8). It currently appears like the data were collected without a clear testable hypotheses in mind and consequentially the author analyses it by throwing all types of combinations of species, sampling method, feeding status and climatic data into a pot and seeing what comes out. As such, the results also seem a little difficult to follow.

Additional comments

Overall this is a fantastic dataset but I feel the information it can provide needs to be told more clearly. I strongly suggest that less is more in the results section. I feel like there is useful information being told here, but even after reading this a number of times, I'm still not quite sure what it is!

The Methods are clear, concise and informative. Minor point: I think there needs to be a little more information on how often and when (for light traps) the data were collected. Were the data collected continuously throughout the full sampling period?
(I particularly like the YouTube clip)

Minor comments/edits are in the attached pdf
Minor comments also copied below:
Line 41: refers to a study identifying a peak in male An. gambiae in exit collections three weeks after rain: Just a point of interest/query: it is odd that this is specifically male mosquitoes - Can the numbers of male mosquitoes in exit traps work as a proxy for population size or are they just looking for optimal environmental conditions?
Line 106: latitude should be 23deg 43’ (currently coords put Furvela in the sea)
Line 170: the equation when applied to the example given (20mm rain over seven days versus 140mm rain over one day) gives the same answer for both so seems contradictory to what is stated – i.e. these different temporal distributions of rainfall are likely to have differing effects.
Line 183: I am not sure/concerned regarding Pearson’s analyses: it is not the best analyses to use on time series data as far as I am aware – this could be causing spurious correlations/’causations’
Line 211: The figures shown don’t correspond to the figures given in Table 1
Lines 234-237: The effect of the LLINs on An. funestus is interesting – why is it disproportionately affecting exit traps. The decline in An. gambiae is also very interesting – it seems to be a increasing phenomenon – often, it seems, just prior to the implementation of a control regime (not really a comment/edit, just interesting :-))
Line 262: only a single Figure 7 was included with the manuscript –
Para 315-322: This all seems a bit odd – why would gravid females be associated with a temperature measure more influential on larval populations? (Possibly due to oddities from Pearson’s on time series?). Also not sure the micro-climate point helps to explain things. The final sentence I really don’t understand at all.

Annotated reviews are not available for download in order to protect the identity of reviewers who chose to remain anonymous.

Reviewer 2 ·

Basic reporting

No comment

Experimental design

Experiment explanation is not clear please add details of malaria situation in the sampling sites and vector control in this area.

1 Light traps: What is the criteria for selecting the houses and placing the traps? Pleas mention in the manuscript.
2 Resting collecting: please more explain in details. How many days of collection? How many people for collection as the whole night? Did they turn over the resting collection in different houses?
3 Did they run as the same time of collection or separately>?

Validity of the findings

This is just the confirmation data about the temperature effecting to mosquito biology. Need to emphasize what is the newthing from this paper.

Additional comments

1. Please check Scientific names and s.l. and in discussion, please add the references,
• Line 39 Change Anopheles gambiae >>> An. gambiae and please check Scientific name and species complex (s.l. ) in the whole document.
• Line 70 Change Anopheles funestus >>> An. funestus
• Line 211 Change Anopheles gambiae >>> An. gambiae
• Line 304 add reference
• Line 348 add reference
• Line 350-1 add reference

·

Basic reporting

The author is a very well-experienced researcher, so this reflects to the overall quality of the manuscript. Think that the scope and the presentation are within the PeerJ interests in Science communication. Although introduction is to me a little bit longer than the ideal length, it reflects the natural history of the malarial vectors studied. One drawback of the introduction is that hypothesis is not explicited, so it seems that the author made a data mining kind-of study (which to me is not the case). Like very much the raw data, it reflects comprehensively data that was obtained in the field. Figures are good, but I would like to see a Figure for the study area. The written English is adequate for a native speaker.

Experimental design

It seems that the present work was not intended to become this manuscript in principle, but it became anyway. That't fine to me because is so hard to get data in the field, specially in África, I guess. But, the problem with this approach is to have biased data to deal with. If it was made by other one, I would reject it, but the present one made a beautiful job, trying to link every pattern with possible underlying biological processes. Notwithstanding, my compliments to mosquito collections carried out, it was really not easy to get this data. And, good to have the raw mosquito data along. Also, liked very much to see exactly how many individuals per Species were tested for DNA Species confirmation. This evidentiates the accuracy of results shown. Ethics considered.

Validity of the findings

Raw data speaks per itself. It was a data-mining study with a set of possible hypotheses that offer explanation to the biological varition observed in the response of malarial vectors to micro-climate variables. The validity of this kind of study is only acchieved if carried out by someone that knows the field and have years on the research avenue. Fortunately, this is the case. Although I have some restrictions to correlation studies as this one, I was convinced by the knowledge and literature search disscussed in the Discussion part. I also agree with the findings about the overall importance of micro-Temperature variations on the mosquito behaviour and frequency. On the other hand, I was surprised to see the lack of importance of rainfall. So, if this is true, we are talking about two Super-vectors for the Global Warming era, when temperature goes up and rainfall goes down.

Additional comments

Congrats. Really enjoyed yourself as a field scientist (I have watched the video, and I invite you to watch mine: https://www.youtube.com/watch?v=di7TRsBuTpk). The work is fine, and to me is a data mining study with potential hypotheses linking pattern to process. Overall, the credibility of your work is yourself, as it is shown on the written knowledge of your ms. Having said that, I believe that the findings are robust, with micro-temperature having a positive effect on African malarial vectors, and rainfal being unimportant. Therefore, I think, maybe, conclusions could be extended a little bit. As most Global Warming scenarions involve both temperature increase and precipitation decrease, this work tells us that these vectors will become more well-adapted in the future, which would diminish the odds of having a malaria elimination program completed in Africa.

---

## Round 0.2 · Minor Revisions

I agree with the reviewer #1 that a better accompanying document to allow others to make use of the raw data should be addressed to the supplemental material.

Please check the annotated pdf attached to make the corrections pointed and update the coordinates as raised by the reviewer.

Reviewer 1 ·

Basic reporting

Not all comments have been addressed for example the coordinates given have not been updated and still fall in the sea. See annotated pdf attached.

The value of this work is the excellent dataset supplied in the supplemental info – some but not all have ‘information’ tabs, and of these not all are very clear.

A distinct explanatory doc/text file accompanying each datasheet to make each column heading nice and clear for anyone hoping to do further analyses would be more useful (though not essential). Examples of things that could be clarifies: in the Funestus haploides raw data, what is the ‘tossed’ column?. In the Mosquito collection: what do ‘Sp A male’? ‘Sp A female’ refer to?

For the Parasitology survey datasheet, my computer didn’t like this and said it detected a ‘problem’ –need to check this.

Experimental design

As before - ok

Validity of the findings

As stated previously, alternative methods of analyses could have been more appropriate, but accept that with the dataset being provided, others can attempt more in depth analyses. Happy with this.

Additional comments

This is a great dataset and I look forward to seeing it published.

Annotated reviews are not available for download in order to protect the identity of reviewers who chose to remain anonymous.

---

## Round 0.3 · accepted · Accept

The author has properly addressed all the comments made by me and the reviewer.